# The Impact of COVID-19 on Gastrointestinal Motility Testing in Asia and Europe

**DOI:** 10.3390/jcm9103189

**Published:** 2020-10-01

**Authors:** Hideki Mori, Jolien Schol, Annelies Geeraerts, I-Hsuan Huang, Sawangpong Jandee, Sutep Gonlachanvit, Ping-Huei Tseng, Ching-Liang Lu, Takeshi Kamiya, Nayoung Kim, Yeong Yeh Lee, Shiko Kuribayashi, Jan Tack, Hidekazu Suzuki

**Affiliations:** 1Department of Chronic Diseases, Metabolism and Ageing, Translational Research Center for Gastrointestinal Diseases (TARGID), University of Leuven, 3000 Leuven, Belgium; hideki.mori@kuleuven.be (H.M.); jolien.schol@kuleuven.be (J.S.); annelies.geeraerts@kuleuven.be (A.G.); alberttsgh@gmail.com (I.-H.H.); tekikung@gmail.com (S.J.); jan.tack@kuleuven.be (J.T.); 2Gastroenterology and Hepatology Unit, Division of Internal Medicine, Faculty of Medicine, Prince of Songkla University, Songkhla, Hatyai 90110, Thailand; 3Center of Excellence on Neurogastroenterology and Motility, Faculty of Medicine, Chulalongkorn University, Bangkok 10330, Thailand; gsutep@hotmail.com; 4Department of Internal Medicine, National Taiwan University Hospital, Taipei 100, Taiwan; pinghuei@ntu.edu.tw; 5Endoscopy Center for Diagnosis and Treatment, Taipei Veterans General Hospital, Taipei 112, Taiwan; cllu@ym.edu.tw; 6Department of Medical Innovation, Nagoya City University Graduate School of Medical Sciences, Nagoya 467-8601, Japan; kamitake@med.nagoya-cu.ac.jp; 7Department of Internal Medicine and Liver Research Institute, Seoul National University College of Medicine, Seoul 03080, Korea; nakim49@snu.ac.kr; 8School of Medical Sciences, Universiti Sains Malaysia, Kota Bharu 15200, Malaysia; justnleeyy@gmail.com; 9Department of Gastroenterology and Hepatology, Gunma University Graduate School of Medicine, Maebashi 371-8511, Japan; shikokuri@yahoo.co.jp; 10Division of Gastroenterology and Hepatology, Department of Internal Medicine, Tokai University School of Medicine, Isehara 259-1193, Japan

**Keywords:** SARS-CoV-2, COVID-19, oesophageal manometry, catheter-based pH-monitoring, wireless pH-monitoring (Bravo^®^), anorectal manometry, breath tests, motility disorders of the gastrointestinal tract, infection prevention

## Abstract

Background: The new coronavirus disease (COVID-19) has high infection and mortality rates, and has become a pandemic. The infection and mortality rates are lower in Asian countries than in European countries. This study aimed to conduct a survey on the effects of COVID-19 on the capacity to perform gastrointestinal motility tests in Asian countries compared with European countries. Methods: We used the questionnaire previously established by our team for researchers in European countries. The correlation between the decreased rate of gastrointestinal motility and function tests, and the infection/mortality rates of COVID-19 and stringency of a government’s interventions in each country was analysed and protective measures were assessed. Results: In total, 58 gastroenterologists/motility experts in Asian countries responded to this survey. The infection/mortality rates of COVID-19 and Stringency Index had a significant impact on the testing capacity of oesophageal manometry and catheter-based pH monitoring. In European countries, most facilities used filtering facepiece 2/3 (FFP2/3) masks during oesophageal motility studies. Meanwhile, in Asian countries, most facilities used surgical masks. Conclusion: The total infection and mortality rates of COVID-19 can affect the rate of gastrointestinal motility testing and the type of protective equipment that must be used.

## 1. Introduction

The new coronavirus disease (COVID-19) has high infection and mortality rates, and has become a pandemic. Potential infection routes of the severe acute respiratory syndrome coronavirus 2 (SARS-CoV-2), leading to COVID-19, include transfer via aerosol and droplets from the nasopharynx, and by physical contact [1]. 

Gastrointestinal motility and function examinations such as oesophageal manometry, catheter-based pH monitoring, wireless pH monitoring (Bravo^®^, Medtronic Inc., Shoreview, MN, USA), anorectal manometry and breath tests are performed to diagnose gastrointestinal motility disorders [2,3,4,5]. Positioning of catheters for gastrointestinal motility tests may induce coughing or sneezing. Hence, there is a significant concern regarding the transmission of this aerosol-borne infection among health care workers [6,7]. Moreover, anorectal manometry can lead to contact infection [8]. 

To date, the number of COVID-19 cases is increasing continuously worldwide. Even in areas where the number of infections has decreased, the risk cannot be completely eliminated. 

Hospitals in several countries postponed non-emergency surgeries and tests during the spread of COVID-19. However, when the number of infections decreased to a sufficient degree, appropriate infection control measures were taken before performing gastrointestinal motility and function tests. Previously, we investigated the impact of the COVID-19 pandemic on the capacity to perform gastrointestinal motility and function tests in European countries [8]. In Asian countries, COVID-19 first spread in China and then in nearby countries [9]. Although the epidemic itself began in Asia, not in Europe, the infection and mortality rates are lower in Asian countries than in European countries [10].

In this study, we conducted a survey to evaluate the effects of COVID-19 on the capacity to perform gastrointestinal motility tests in Asian countries compared with European countries. Moreover, we analysed how the infection and mortality rates of COVID-19 in each country affected the capacity to perform gastrointestinal motility tests in Asian and European countries, and which measures were taken to protect patients and staff from SARS-CoV-2. 

## 2. Methods

### 2.1. Survey Using Questionnaires Established for Centres in Asian Countries

To assess the impact of the pandemic on the capacity to perform motility and function tests in Asian countries, we have used the 26-item questionnaire previously established by our team of researchers in the European countries [11]. We requested the motility unit members of the Asian Neurogastroenterology and Motility Association to participate in this survey. 

In this study, the following gastrointestinal motility procedures were assessed in the questionnaire: oesophageal manometry, catheter-based pH monitoring, wireless pH monitoring (Bravo^®^), anorectal manometry and breath tests. Specifically, the questions aimed to evaluate whether the centre reduced or discontinued performing motility and functional tests (particularly at which time point in the pandemic and to what extent). Moreover, the questions assessed the timing for restarting these activities and the capacity to which they were restarted. In addition, the questions evaluated the protective equipment and screening methods applied by the respective centres during the restarting period of these tests.

Findings from Asian countries were compared to their European counterparts [11]. The European centres represent the following countries: Belgium (*n* = 1); France (*n* = 3), Germany (*n* = 4), Spain (*n* = 3), Israel (*n* = 2), Portugal (*n* = 3), Denmark (*n* = 1), Turkey (*n* = 2), Italy (*n* = 3), UK (*n* = 1), Ireland (*n* = 1), Poland (*n* = 1), Romania (*n* = 3), Croatia (*n* = 1), Russia (*n* = 1) and Switzerland (*n* = 1) [11]. The response rate was 88.6% [11].

### 2.2. Analysis of the Relationship between the Reduced Rate of Gastrointestinal Motility and Function Tests

Data on COVID-19 infection and the mortality rates in each country were obtained from Worldometers (http://www.worldometers.info), The Real Time Statistics Project, as of the 12 June 2020 (Appendix A). Data on the stringency of government interventions in each country were obtained from the Covid-19 Government Response Stringency Index (Stringency Index) developed by researchers at Oxford University (https://www.bsg.ox.ac.uk/research/research-projects/coronavirus-government-response-tracker), which is based on nine indicators: school closures; workplace closures; cancellation of public events; restrictions on public gatherings; closures of public transport; stay-at-home requirements; public information campaigns; restrictions on internal movements; and international travel controls, rescaled to a value from 0 to 100 (100 = strictest), as of the 12 June 2020 (Appendix A).

### 2.3. Statistical Analysis

Unless otherwise specified, numerical data were presented as median (range). The correlation between the reduced rate of gastrointestinal motility tests and the infection or mortality rates of COVID-19 was analysed using Spearman’s rank correlation. The differences in personal protective equipment used for different gastrointestinal motility examinations and screening procedures prior to performing gastrointestinal motility tests were compared using Fisher’s exact test. Statistical analyses were performed using the Statistical Package for the Social Sciences software for Windows version 25 (SPSS Inc., Chicago, IL, USA). 

## 3. Results

### 3.1. Impact of the Build-Up and Peak Period of the COVID-19 Pandemic in Asian and European Countries

#### 3.1.1. Characteristics of Participating Centres

We conducted a survey to evaluate the impact of COVID-19 on gastrointestinal motility testing in hospital settings. In total, 58 gastroenterologists/motility experts belonging to tertiary referral centres in Asia responded to this survey. The response rate is 87.8%. The centres are located in the following countries: Mainland China (*n* = 1), Hong Kong (*n* = 1), Japan (*n* = 20), Korea (*n* = 7), Malaysia (*n* = 6), Philippines (*n* = 1), Singapore (*n* = 1), Taiwan (*n* = 13) and Thailand (*n* = 8). 

#### 3.1.2. Impact on Oesophageal Manometry

In short, COVID-19 has a significant impact on the capacity and extent to which oesophageal manometry can be performed. In total, 36 of 50 performing centres reduced or stopped performing this test on average on the 22 March 2020 (range 20 January 2020–20 April 2020). These centres reduced their capacity by 90% (range 12–100%). Moreover, 13 centres immediately stopped their activities, and two centres gradually reduced their activities before completely stopping. In total, 21 centres reduced their activities by 80% (range 12–99%). However, they still performed urgent oesophageal manometry; for example, for functional severe dysphagia with weight loss and/or risk of aspiration, for oesophageal manometry prior to treatment for achalasia with major impact, in order to assess the manometric pattern of the disease, for non-cardiac chest pain with high impact on quality of life (QoL), and also for refractory oesophageal symptoms with weight loss, persistent regurgitation, risk of aspiration, and/or high impact on QoL. The centres that reduced or stopped their testing resumed after a median of 73 (28–122) days. When analysed according to country, the centres in three of eight countries in Asia and 12 of 19 countries in Europe completely discontinued their tests (37.5% and 63.2%, respectively) [11]. The relationships between the median reduction rates of oesophageal manometry, total infection rates and mortality rates of COVID-19, and the Stringency Index, are shown in Figure 1, respectively. There was a significant trend between the infection rates of COVID-19, the Stringency Index and the median reduction rates of oesophageal manometry (r = 0.431, *p* = 0.025; Figure 1A, r = 0.502, *p* = 0.008; Figure 1C). There was a marginally significant trend between the mortality rates and the median reduction rates of oesophageal manometry (r = 0.352, *p* = 0.072; Figure 1B).

#### 3.1.3. Impact on Catheter-Based pH Monitoring

COVID-19 also affected the capacity and extent to which catheter-based pH monitoring could be performed. Briefly, 35 of 49 centres had reduced or stopped this test on average on the 23 March 2020 (range 20 January 2020–17 May 2020). Eighteen centres immediately stopped their activities, and two centres gradually reduced their activities before completely stopping. Fifteen centres reduced their activities by 66% (range 20–95%). The centres that reduced or stopped their tests resumed after a median of 73 (28–122) days. When analysed according to country, the centres in two of seven countries in Asia and 14 of 19 countries in Europe completely stopped their tests (28.6% and 73.7%, respectively) [11]. The relationships between the median reduction rates of catheter-based pH monitoring, total infection rates and mortality rates of COVID-19, and the Stringency Index, are presented in Figure 2, respectively. There was a significant trend between the infection rates and mortality rates of COVID-19, the Stringency Index and the median reduction rates of catheter-based pH monitoring (r = 0.565, *p* = 0.003; Figure 2A, r = 0.501, *p* = 0.009; Figure 2B, r = 0.475, *p* = 0.014; Figure 2C). Most countries in Europe, particularly those with SARS-CoV-2 infection rates above 3000 per million people, had discontinued performing catheter-based pH monitoring. 

#### 3.1.4. Impact on Wireless pH Testing (Bravo^®^)

Similarly, COVID-19 had a significant impact on wireless pH testing (Bravo^®^). Only 13 centres performed this type of test before the start of the pandemic, and eight centres had reduced or stopped performing this test on average on the 26 March 2020 (range 7 February 2020–1 April 2020). Eight centres reduced their capacity by 100% (range 80–100%). Five centres immediately stopped their activities, and only one centre did not completely discontinue the Bravo^®^ test (80%). The centres that reduced or stopped this test resumed after a median of 68 (49–82) days. Notably, among the centres in Taiwan, four out of five did not reduce or stop performing this test. The relationships between the median reduction rates of wireless pH testing, total infection rates, and mortality rates of COVID-19, and the Stringency Index, are depicted in Figure 3, respectively. In both Asia and Europe, all countries, except Taiwan, had almost completely discontinued this type of testing.

#### 3.1.5. Impact on Anorectal Manometry

Anorectal manometry was performed in 24 of 58 centres before the start of the COVID-19 pandemic. However, 21 centres completely discontinued their activity (range 31–100%) during the pandemic. Three centres did not reduce their rate of performing anal manometry testing. Nevertheless, 16 centres immediately reduced their capacity to perform the test, and five gradually reduced their capacity before completely stopping. Eight centres reduced their activities by 50% (31–80%). The median time of reduction or stopping anal manometry was on average on the 18 March 2020 (range 20 January 2020–20 April 2020). The centres that reduced or stopped their activities resumed after a median of 75 (30–122) days. The relationships between the median reduction rates of anal manometry, total infection rates and mortality rates of COVID-19, and the Stringency Index, are depicted in Figure 4, respectively. No significant trends were observed (Figure 4A–C). Meanwhile, in Europe, anal manometry was stopped in countries with SARS-CoV-2 infection rates above 3000 per million people (Figure 4A) or with mortality rates above 200 per million people (Figure 4B). 

#### 3.1.6. Impact on Breath Tests

Of 58 centres, 37 performed breath tests before the start of the COVID-19 pandemic. However, 23 centres reduced their capacity by 90% (range 21–100%) on average on the 2 March 2020 (range 20 January 2020–8 April 2020). Fourteen centres did not change their capacity in performing breath tests due to COVID-19. However, the capacity of conducting this test was immediately reduced in 18 centres, and was gradually reduced before completely stopping all activities in five centres. Three centres reduced their activities by 50% (range 21–90%). The centres that reduced or stopped their capacity resumed after a median of 83 (30–121) days. No significant trends were observed in the relationships between the median reduction rates of breath tests, total infection rates, mortality rates of COVID-19 and Stringency Index (Figure 5A–C).

### 3.2. Planned Use of Personal Protective Equipment for Motility and Function Testing during the Early Recovery Phase of the COVID-19 Pandemic 

The most commonly used protective equipment during oesophageal manometry, catheter-based pH monitoring and wireless pH monitoring (Bravo^®^) were surgical masks, face shields, water-resistant gowns and standard gloves (Table 1). In European countries, filtering facepiece 2 (FFP2) masks are more frequently used than surgical masks, and significant differences were observed between Asian and European countries (Oesophageal manometry; *p* < 0.001, Catheter-based pH-monitoring; *p* < 0.001, Bravo^®^ pH-capsule; *p* = 0.005, Anal manometry; *p* < 0.001, Breath tests; *p* = 0.001). Hairnets were used more often in European countries than in Asian countries during oesophageal manometry and catheter-based pH monitoring (Oesophageal manometry; *p* = 0.010, Catheter-based pH-monitoring; *p* = 0.001). 

Anamnestic risk assessments and temperature checks were the most common screening procedures that are and can be used by the centres to detect SARS-CoV-2 infection prior to performing investigations (Table 2). More than 90% of institutions in Europe perform anamnestic risk assessments, but they were only used in about 70% in Asian countries. Polymerase chain reaction (PCR) was the technique typically used to diagnose acute COVID-19 in 20–29% of institutions at each motility centre in Asian countries. The rate was evidently lower than that of European countries (Table 2).

## 4. Discussion

In this study, we have evaluated and compared the impact of COVID-19 on capacity and extent in performing gastrointestinal motility and functional tests in Asian and European countries using the same survey in European countries [8]. 

The factors of performing or not performing the motility tests are dependent on the urgency of the examination, risk of COVID-19 associated with the examination among examiners, infection and mortality rates and stringency of government interventions in each country. Moreover, the other potential factors, such as the demographic profile of the patients, socioeconomics including the cost of the testing, healthcare systems’ need to save the personal protective equipment and prepare for the pandemic, patient preferences, and hospital policies, might have impacts on the reduction in gastrointestinal motility studies. 

Oesophageal manometry and catheter-based pH monitoring are important preoperative tests for oesophageal achalasia and refractory gastroesophageal reflux disease [4,12,13,14]. In addition, anal manometry is performed pre-operatively for certain anal conditions, and if not performed it may affect postoperative risk [15]. Delayed testing results in a medical and social disadvantage that prevents patients from receiving appropriate treatment. 

Positioning of oesophageal catheters for gastrointestinal motility and function tests may cause coughing or sneezing. Hence, there is a concern for the risk of transmitting aerosol-borne infection to health care workers [6,7]. The rate of viral transmission via aerosol is likely high when people are exposed to a great concentration of contaminated aerosol in a closed space, such as an examination room, for a certain period of time. In relation to these reasons, tests using an oesophageal catheter are associated with a relatively high risk of COVID-19 among examiners. However, the number of droplets generated during catheter insertion is not known. Thus, a thorough evaluation must be conducted in the future as the assessment can provide a quantitative understanding of the risk associated with catheter insertion among examiners. In addition, both oesophageal and anal catheters can be contaminated with body fluids. SARS-CoV-2 has a high risk of contact infections [1]. These tests carry the risk of contact infections. Hence, proper protective measures must be taken to prevent infection.

Although the reason is not yet fully elucidated, the infection and mortality rates of COVID-19 differ significantly between Asia and Europe [10] (Appendix A). In Europe, the rapid spread of COVID-19 has resulted in high infection and mortality rates. Therefore, health workers often faced extreme stress due to increased demands on health and social care system [16]. In addition, lockdown policies were adopted in most European countries, which is reflected in the Stringency Index (Appendix A). Thus, the limited number of non-urgent visits to hospitals have affected the rate of gastrointestinal motility and function tests [17]. Furthermore, due to the fear of the pandemic, many patients prefer not to get tested. By contrast, in Asian countries, the overall infection and mortality rates remained relatively low compared with those of European countries, and the prior risk of SARS-CoV-2 infection among asymptomatic patients may be relatively lower. 

Interestingly, our study has showed that the infection/mortality rates of COVID-19 and the Stringency Index had a more significant impact on the testing capacity of oesophageal manometry and catheter-based pH monitoring (Figure 1 and Figure 2). In contrast, no correlation was shown for wireless pH testing, anorectal manometry and breath tests (Figure 3, Figure 4 and Figure 5). 

These results imply that the potential risk of SARS-CoV-2 infection from the testing procedures of oesophageal manometry and catheter-based pH monitoring is high, as is the importance of protecting examiners from droplets during insertion and catheter removal. Wireless pH testing has the same risk, however no correlation was shown for wireless pH testing because it was completely discontinued in almost all countries except Taiwan. On the other hand, anal manometry and breath tests are assumed to have a low risk of droplet infection associated with cough reflex and catheter removal, therefore these tests may not have been discontinued even in countries with high infection rates of SARS-CoV-2.

The significant difference in selection of personal protective equipment between Asian and European countries is also noteworthy (Table 1). The guidelines for gastrointestinal motility and function tests in Europe and Asia recommend the use of full personal protective equipment, including N95 masks or equivalent filtering facepiece respirators, hairnets/hoods, isolation gowns, double gloves, goggles or face shields when treating patients who are at high risk of COVID-19 or those whose risk is unknown and standard equipment, such as surgical masks, isolation gowns, gloves and hairnets/hoods when managing low-risk patients [11,18]. While the concept of infection prevention during gastrointestinal motility and function tests is similar between Europe and Asia, our survey showed a major difference in the selection of personal protective equipment between Asian and European countries (Table 1). In European countries, most facilities use FFP2 or FFP3 masks during oesophageal manometry, catheter-based pH monitoring, and wireless pH monitoring (Bravo^®^). Meanwhile, in Asian countries, most facilities use surgical masks. In addition, a higher number of healthcare professionals use hairnets during oesophageal manometry and catheter-based pH monitoring tests in Europe. These findings show that FFP2/FFP3 masks are more commonly used in Europe and this reflects the high infectious risk of asymptomatic and pre-symptomatic COVID-19. There is still no evidence that a surgical mask is sufficient for protection from SARS-CoV-2 infection especially when performing oesophageal manometry and catheter-based pH monitoring, thus our study has shown that the scientific evidence should be established, and the guidelines need to be modified according to the evidence. 

In addition, most facilities in Europe conduct an anamnestic risk assessment before performing gastrointestinal motility and function tests. However, in Asia, the rate of this assessment was only done in 66–79% of centres (Table 2). Reverse transcription polymerase chain reaction (RT-PCR) using nasopharyngeal swabs is commonly performed to diagnose SARS-CoV-2 infection in Europe. Since the RT-PCR test kits for SARS-CoV-2 are limited and the test is expensive, therefore environmental factors, such as infection rates in nearby countries, are taken into account instead. Moreover, an anamnestic risk assessment and temperature check should be performed in countries, territories, or areas reported with active COVID-19 cases [11]. 

## 5. Conclusions

The infection and mortality rates of COVID-19 have significantly affected the capacity and extent of performing gastrointestinal motility tests and also the type of protective equipment use. Whether these tests should be performed or postponed and the appropriate protective strategies that must be used against COVID-19 may be determined through consideration of SARS-CoV-2 infection rates, the urgency of the tests and the risks associated with the procedures among examiners.

## Figures and Tables

**Figure 1 jcm-09-03189-f001:**
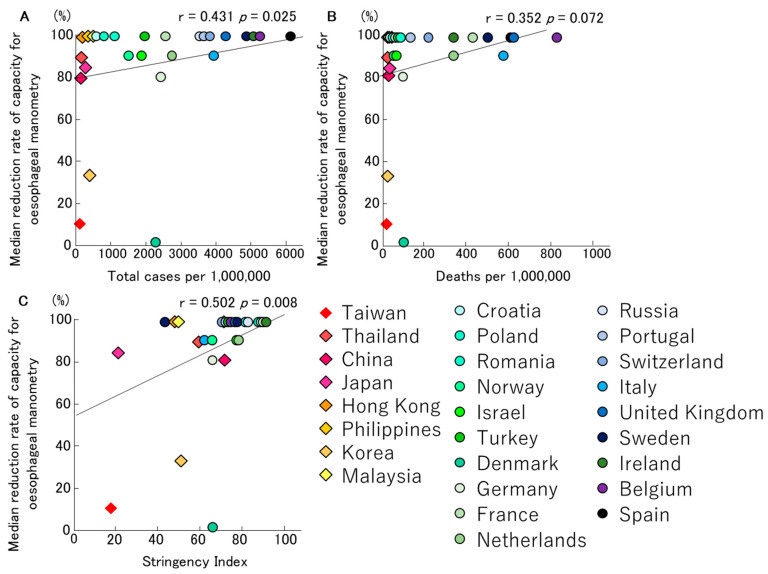
The relationships between median reduction rates of capacity for oesophageal manometry, total infection rate (**A**)/mortality rate (**B**) for COVID-19 and Stringency Index (**C**) in Asian and European countries, are described. There was a significant trend between the infection rates of COVID-19, the Stringency Index and the median reduction rates of oesophageal manometry.

**Figure 2 jcm-09-03189-f002:**
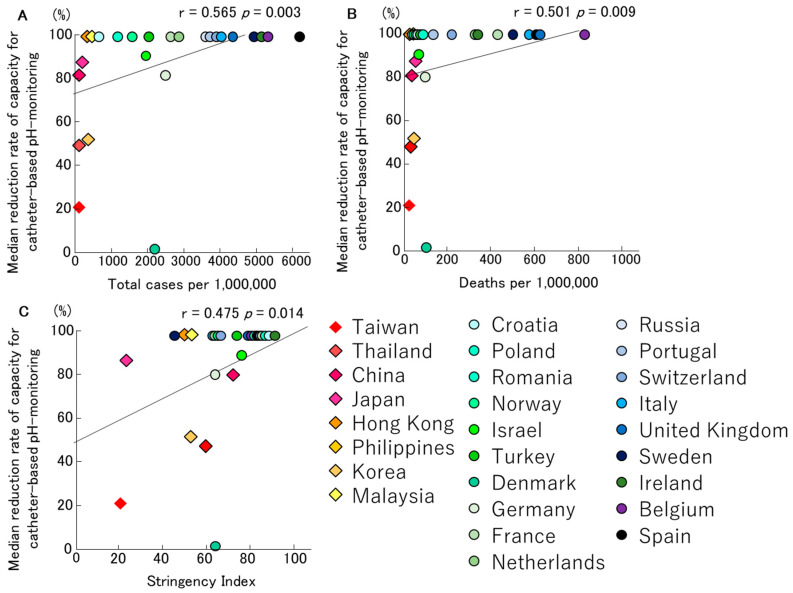
The relationships between median reduction rates of capacity for catheter-based pH monitoring, total infection rate (**A**)/mortality rate (**B**) for COVID-19 and Stringency Index (**C**) in Asian and European countries, are described. There was a significant trend between the infection rates and mortality rates of COVID-19, the Stringency Index and the median reduction rates of catheter-based pH monitoring.

**Figure 3 jcm-09-03189-f003:**
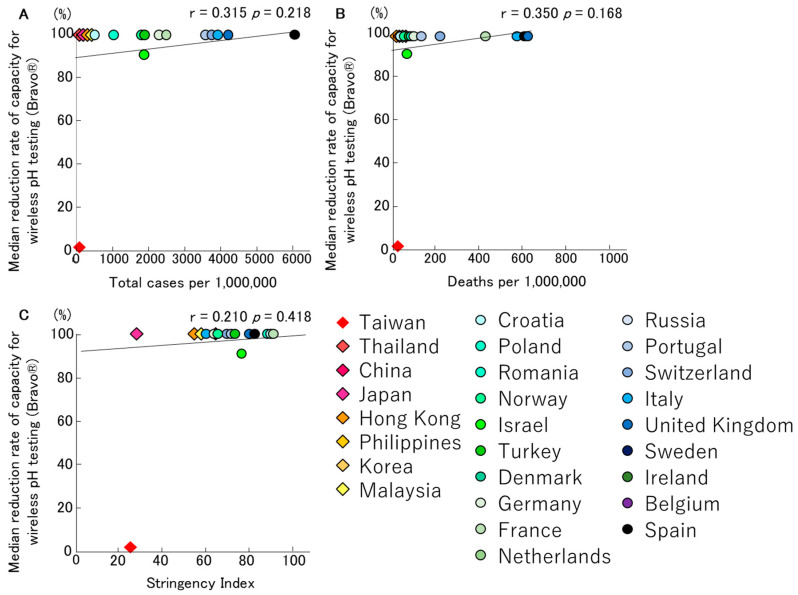
The relationships between median reduction rates of capacity for wireless pH testing (Bravo^®^), total infection rate (**A**)/mortality rate (**B**) for COVID-19 and Stringency Index (**C**) in Asian and European countries, are described. Almost all countries, except Taiwan, had completely discontinued this type of testing.

**Figure 4 jcm-09-03189-f004:**
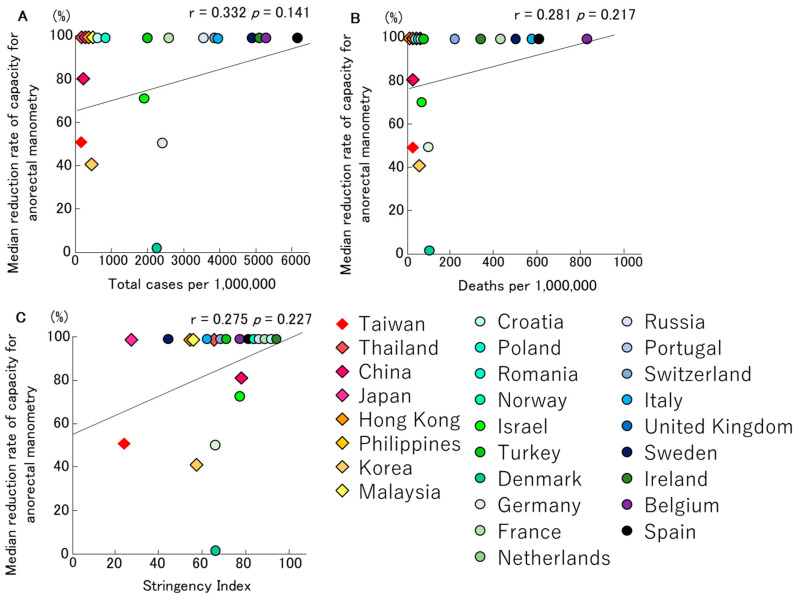
The relationships between median reduction rates of capacity for anorectal manometry, total infection rate (**A**)/mortality rate (**B**) for COVID-19 and Stringency Index (**C**) in Asian and European countries, are described. No significant trends were observed. Meanwhile, in Europe, anal manometry was stopped in countries with SARS-CoV-2 infection rates above 3000 per million people or with mortality rates above 200 per million people.

**Figure 5 jcm-09-03189-f005:**
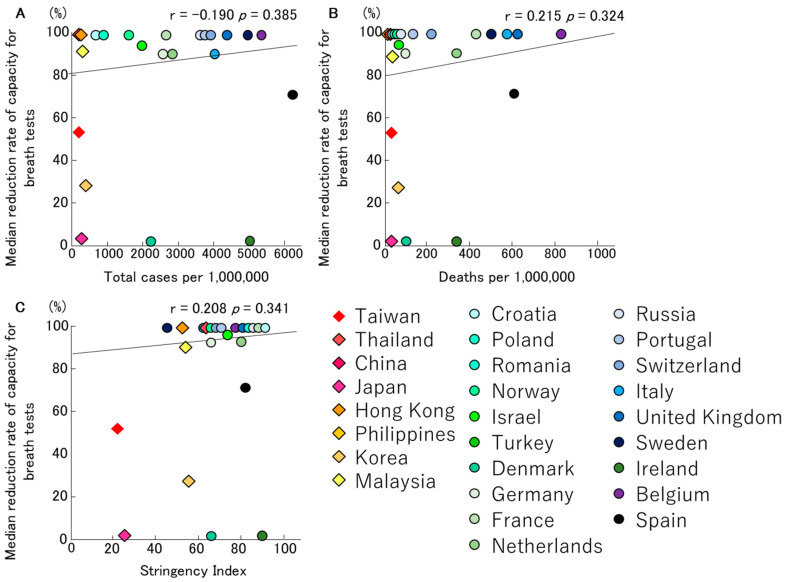
The relationships between median reduction rates of capacity for breath tests, total infection rate (**A**)/mortality rate (**B**) for COVID-19 and Stringency Index (**C**) in Asian and European countries, are described. No significant trends were observed in the relationships between the median reduction rates of breath tests, total infection rate/mortality rate of COVID-19 and Stringency Index.

**Table 1 jcm-09-03189-t001:** Personal protective equipment for different gastrointestinal motility investigations.

	Oesophageal Manometry			Catheter-Based pH-Monitoring			Bravo^®^ pH-Capsule			Anal Manometry			Breath Tests		
Protection Mechanism	AS (*n* = 49)	EU (*n* = 30)	*p*-Value	AS (*n* = 50)	EU (*n* = 31)	*p*-Value	AS (*n* = 12)	EU (*n* = 13)	*p*-Value	AS (*n* = 25)	EU (*n* = 29)	*p*-Value	AS (*n* = 35)	EU (*n* = 22)	*p*-Value
None	2%	0%	1.000	2%	0%	1.000	8%	0%	0.48	4%	0%	0.463	20%	0%	0.025
Negative pressure room	2%	3%	1.000	2%	3%	1.000	0%	8%	1.000	0%	3%	1.000	3%	0%	1.000
Surgical mask	82%	27%	**0.000**	80%	26%	**0.000**	83%	15%	1.000	88%	55%	**0.015**	69%	32%	**0.013**
FFP2-mask	14%	73%	**0.000**	14%	71%	**0.000**	17%	77%	**0.005**	4%	55%	**0.000**	9%	50%	**0.001**
FFP3-mask	4%	20%	**0.048**	4%	23%	**0.024**	8%	23%	0.593	4%	10%	0.615	3%	23%	**0.028**
Goggles	43%	43%	1.000	42%	45%	0.821	42%	31%	0.688	36%	34%	1.000	31%	45%	0.398
Face shield	63%	80%	0.137	62%	81%	0.089	83%	77%	1.000	48%	52%	0.79	49%	68%	0.178
Hairnet	45%	77%	**0.010**	44%	81%	**0.001**				36%	66%	0.055	63%	64%	0.062
Water-resistant gown	58%	77%	0.411							64%	65%	1.000	49%	55%	0.787
Non-water-resistant gown	63%	73%	0.461	62%	74%	0.334	83%	85%	0.593	12%	28%	0.191	91%	23%	0.239
Long sleeved gloves	16%	23%	0.557	16%	23%	0.559	8%	15%	1.000	8%	21%	0.262	94%	0%	0.281
Standard gloves	10%	23%	0.195	10%	23%	0.197	8%	23%	0.593	52%	83%	0.431	26%	82%	0.747
Overshoe covers	90%	77%	0.195	88%	81%	0.521	25%	69%	1.000	0%	3%	1.000	0%	0%	1.000

Abbreviations: FFP = Filtering Face Piece, AS = Asian countries, EU = European countries. Bold value indicates a significant difference.

**Table 2 jcm-09-03189-t002:** Screening procedures used prior to performing gastrointestinal motility investigations.

Screening Procedure	Oesophageal Manometry			Catheter-Based pH-Monitoring			Bravo^®^ pH-Capsule			Anal Manometry			Breath Tests		
	AS (*n* = 50)	EU (*n* = 30)	*p*-Value	AS (*n* = 49)	EU (*n* = 31)	*p*-Value	AS (*n* = 14)	EU (*n* = 13)	*p*-Value	AS (*n* = 24)	EU (*n* = 29)	*p*-Value	AS (*n* = 35)	EU (*n* = 22)	*p*-Value
None	10%	0%	0.151	8%	0%	0.154	7%	0%	1.000	4%	0%	0.452	14%	0%	0.145
Anamnestic risk assessment	68%	93%	**0.011**	71%	94%	**0.021**	79%	92%	0.595	71%	90%	0.156	66%	100%	**0.002**
Temperature check	84%	83%	1.000	82%	84%	1.000	86%	77%	0.648	88%	83%	0.715	74%	82%	0.746
Nasopharyngeal PCR-swab	24%	50%	0.027	20%	48%	**0.013**	29%	46%	0.440	25%	38%	0.383	20%	36%	0.221
CT-scan	2%	3%	0.612	4%	3%	1.000	7%	0%	1.000	0%	3%	1.000	0%	5%	0.386
Serology test	0%	7%	0.137	0%	10%	0.055	0%	8%	0.481	0%	7%	0.494	0%	9%	0.144
Saturation O_2_	0%	3%	0.375	0%	3%	0.388	0%	0%	1.000	0%	3%	1.000	0%	5%	0.386

Abbreviations: PCR, polymerase chain reaction; CT, computed tomography; AS, Asian countries; EU, European countries. Bold value indicates a significant difference.

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
