# Peer review of "The Impact of COVID-19 on Gastrointestinal Motility Testing in Asia and Europe"

_jcm, 2020, doi:10.3390/jcm9103189_

Round 1

Reviewer 1 Report

Comments→

  1. Authors note that their aim to conduct the effects of COVID-19 on the capacity to perform GI motility testing in Asian countries vs. European countries. Author report this aim with a hypothesis on infection rate and mortality between Asian and European countries. However, this should be based on many more factors such as the demographic profile of the patients, socioeconomics including the cost of the testing, healthcare systems need to save the PPE, and prepare for the pandemic, patient preferences, and hospital policies. It is generally accepted that with the pandemic/ pre-pandemic times, the need for non-urgent or elective procedures go down as recommended by various GI societies. Naturally, GI motility studies are non-emergent procedures which get affected. The findings of this study is in line with this hypothesis. As authors consider only the rate of infection and PPE as the major factors in this process, results and conclusions might not be adequate or cover the whole picture. 
  2. Nevertheless, this study provides insights into the ways the pandemic is affecting the GI motility. 
  3. The authors noted that 58 experts responded to the survey (Asia). What is the rate of responses? How many individuals were contacted of which 58 responded? What is the number in Europe?. Furthermore, are all these centers' tertiary referral centers. Demographics of the centers might play a role (small centers like not to perform) etc. 
  4. Any specific reason for including only Asia and European cohorts? Why not US cohorts?
  5. Authors noted some centers performed “urgent” esophageal manometry (line 116). Please clarify what constitutes urgent manometry.
  6. A table comparing pooled data on Asia and Europe (with different motility studies) and their reduction makes it easy for the audience to grasp. The authors should consider this.
  7. Table 1 interpretation is limited due to small numbers. The type of PPE is dependent again on the availability, preferences of the center. 
  8. Line 222: Authors should also mention various other factors that might play a role in motility testing during the pandemic and post-pandemic times. 
  9. Line 252: Authors should elaborate on the potential reasons for low rates of motility testing→ Specifically due to the fear of the pandemic, many patients do not prefer to get tested. Furthermore, due to the public health officials push for avoiding non-urgent procedures, many patients are not interested to show up for these procedures. 

Reviewer 2 Report

Manuscript is correct from professional point of view, but it is too descriptive. It show how personal protection and number of procedures looks in situation of low or high epidemic risk of COVID, what was already regulated by local/national regulations or guidelines.

In my opinion it wan`t bring conclusion which change anything in general knowledge about endoscopy vs COVID situation.

Reviewer 3 Report

This manuscript describes behaviour of Gastroenterologists operating in Neurogastroenterology Units of Eastern Countries in the COVID-19 era. The study was conducted by administrating the same questionnaire adopted in European Countries, so that a comparison was made between the these two parts of the World. Unlike what seen in Europe, a correlation was found in Eastern Countries between the decrease of catheter-based pH-monitoring  and the severity of infection in the population. Interestingly such a correlation was not found in terms of other functional tests including esophageal manometry, Bravo procedures, ano-rectal manometry, and breath tests. The paper is interesting, but lacks indepth in the evaluation of results. Specifically:

  1. Differences between Asia and Europe may be due to different infectious rates, as suggested in the manuscript, but if this is the case a sub-analysis is needed to compare behaviours in areas with similar rates at the time of the survey.
  2. Regardless of comparisons with European attitudes, the reason to decrease only pH-monitoring according to infectious rates, but no other testing is difficult to understand and should be discussed and clarified for the reader.

Round 2

Reviewer 2 Report

After revision I accept new form of article